# LongVU: Spatiotemporal Adaptive Compression for Long Video-Language Understanding

## Abstract

Multimodal Large Language Models (MLLMs) have shown promising progress in understanding and analyzing video content. However, processing long videos remains a significant challenge constrained by the limited context length. To address this limitation, we propose **LongVU**, a spatiotemporal adaptive compression mechanism to reduce the number of video tokens while preserving visual details of long videos. Our idea is based on leveraging cross-modal query and inter-frame dependencies to adaptively reduce temporal and spatial redundancy in videos. Specifically, we leverage DINOv2 features to remove redundant frames that exhibit high similarity. Then we utilize text-guided cross-modal query for selective frame feature reduction. Further, we perform spatial token reduction across frames based on their temporal dependencies. Our adaptive compression strategy effectively processes a large number of frames with little visual information loss within limited context length. Our LongVU consistently surpass existing methods across a variety of video understanding benchmarks, especially on hour-long video understanding tasks such as VideoMME and MLVU. Given a light-weight LLM, our LongVU also scales effectively into a smaller size with state-of-the-art video understanding performance. Our code will be made publicly available.

## 1 Introduction

Large Language Models (LLMs) (Brown, 2020; Ouyang et al., 2022; OpenAI, 2022; Achiam et al., 2023; Chiang et al., 2023; Touvron et al., 2023; Jiang et al., 2024) manifest universal capabilities that are instrumental in our progress towards general intelligence. Through the integration of modality alignment and visual instruction tuning, Multimodal Large Language Models (MLLMs) (Alayrac et al., 2022; Li et al., 2023b; Zhu et al., 2023; Liu et al., 2024c; Ye et al., 2023; Bai et al., 2023; Chen et al., 2023c; Dong et al., 2024) have demonstrated exceptional competencies in tasks such as captioning and visual question-answering. Recent literatures have initiated explorations of extending MLLMs for the comprehension of video content (Li et al., 2023c; Zhang et al., 2023; Maaz et al., 2023a; Lin et al., 2023; Wang et al., 2024; Liu et al., 2024a). Despite exhibiting potentials across specific benchmarks, effectively processing and understanding of exceedingly lengthy videos remains a significant challenge.

One primary reason is that it is impractical to process all the information for hour-long videos, given that advanced MLLMs represent a single image using hundreds of tokens. For instance, $576 \sim 2,880$ tokens per image are used in LLaVA-1.6 (Liu et al., 2024b) and 7,290 tokens are used in LLaVA-OneVision (Li et al., 2024a). However, a commonly used and computationally manageable context length for multimodal training is 8k, which limits processing 125 frames (2-minutes video) even at 64 tokens per frame, while an hour-long video could require over 200k tokens. Consequently, in video scenarios with an extra temporal dimension, it is intractable for training due to the demand of excessive GPU memory. Various studies have attempted to establish a balance between the number of tokens and the frequency of frame sampling. Most of these studies (Li et al., 2024a; Cheng et al., 2024; Zhang et al., 2024b; Chen et al., 2024) opt for a uniform sampling of a fixed number of video frames as the input. However, these methods naively overlook non-uniform content, e.g., static vs dynamic scenes within the video, as shown in Figure 1. Other approaches (Li et al., 2023c;d; Jin et al., 2023) employ intensive resampling modules that significantly decrease the quantity of visual tokens, leading to a considerable loss of essential visual information.

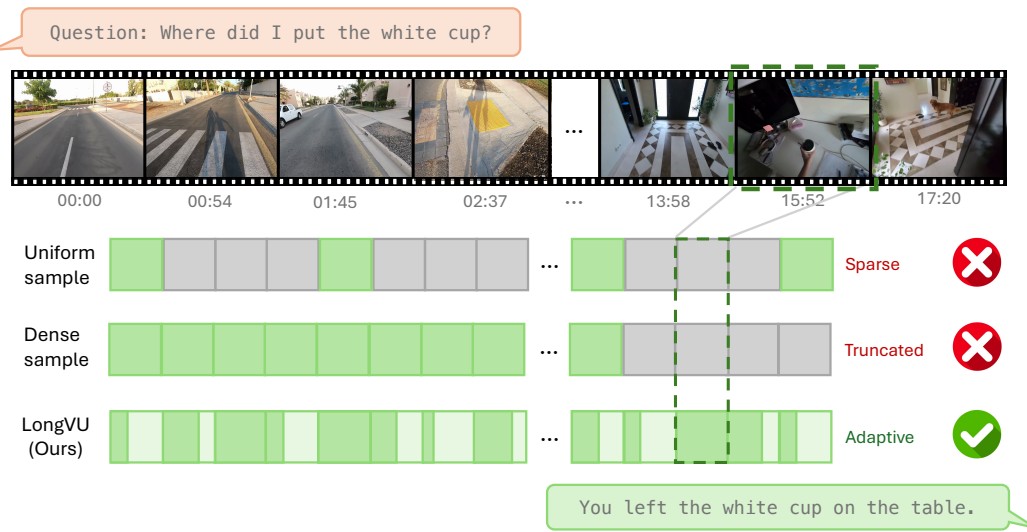

Figure 1: Effectiveness of our LongVU over commonly-used uniform sampling and dense sampling. Uniform sampling overlooks critical frames due to its sparse nature. Dense sampling may surpass the maximum context length, leading to truncation of tokens from targeted frames. In contrast, our method can adaptively conduct spatiotemporal compression, accommodating long video sequences while preserving more visual details.

In this paper, we propose LongVU that aims to preserve as much frame information as possible while accommodating lengthy videos without exceeding the context length of commonly used LLMs. Video by its nature contains significant temporal redundancy. MovieChat (Song et al., 2024) employs a similarity-based frame-level feature selection using visual representation from CLIP (Radford et al., 2021). While we argue that DINOv2 (Oquab et al., 2023), through self-supervised training with a feature similarity objective on vision-centric tasks, captures subtle frame differences and low-level visual features more effectively than vision-language contrastive methods (Radford et al., 2021; Zhai et al., 2023), as shown in Figure 6. Hence, **(1)** we apply a temporal reduction strategy on the frame sequence by leveraging similarity from DINOv2 (Oquab et al., 2023) features to remove redundant video frames. In addition, **(2)** we jointly capture the detailed spatial semantic and long-range temporal context by performing selective feature reduction via cross-modal query, where we preserve full tokens for frames that are relevant to the given text query, while applying spatial pooling to reduce the remaining frames to a low-resolution token representation. **(3)** A spatial token reduction mechanism based on temporal dependencies is applied for excessively long videos. As a result, our model is capable of processing 1fps sampled video input with high performance, which can adaptively reduce the number of tokens per frame to 2 on average to accommodate an hour-long video for MLLM within 8k context length.

To evaluate our method, we conduct extensive experiments across various video understanding benchmarks, including EgoSchema (Mangalam et al., 2024), MVBench (Li et al., 2024b), VideoMME (Fu et al., 2024), and MLVU (Zhou et al., 2024). Our LongVU significantly outperformes several recent open-source video LLM models, such as VideoChat2 (Li et al., 2024b), LongVA (Zhang et al., 2024a), and LLaVA-OneVision (Li et al., 2024a), by a large margin. For example, our LongVU outperforms a strong open-source baseline, LLaVA-OneVision (Li et al., 2024a) by approximately ∼5% in average accuracy. We also observed that our light-weight LongVU, basing Llama3.2-3B (Llama, 2024) as the language backbone, significantly improves over previous state-of-the-art small video-LLMs, e.g., Phi-3.5-vision-instruct-4B (Abdin et al., 2024), by 3.4% on VideoMME Long subset. Our LongVU established new state-of-the-art results on video understanding benchmarks among video-language models. We believe that our proposed approach marks a meaningful progression towards long video understanding MLLMs.

## 2 RELATED WORK

### 2.1 VISION LANGUAGE MODELS

Early visual language models (VLMs) such as CLIP (Radford et al., 2021), is trained with a contrastive loss to project both vision and language embeddings to a shared representation space. SigLIP (Zhai et al., 2023) takes a sigmoid loss instead, allowing further scaling up training batch size with better performance.

The development of LLMs has significantly advanced VLMs. Kosmos-1 (Huang et al., 2023; Peng et al., 2023) introduces an end-to-end framework that integrates visual inputs with LLM in a cohesive training regime. Flamingo (Alayrac et al., 2022) and BLIP-2 (Li et al., 2023a) merge visual and linguistic features through cross-attention and a Q-Former module, respectively. MiniGPT-4 (Zhu et al., 2023) and LLaVA (Liu et al., 2024c) simplify the integration by projecting visual features directly into the LLM embedding space using a MLP.

Later studies (Chen et al., 2023b; Peng et al., 2023; Wang et al., 2023; Chen et al., 2023a) have expanded LMM applications to broader multi-modal tasks, enhancing spatial perception through visual grounding. Recent efforts (Liu et al., 2024b; Dong et al., 2024) aim to create general models that unify diverse tasks, employing sophisticated optimization techniques, high-quality multi-task datasets, and complex training strategies to boost performance across extensive vision-language tasks. Cambrian (Tong et al., 2024) combines features from multiple vision encoders with Spatial Vision Aggregator (SVA) for a more capable MLLM. By exploring different vision encoders, Cambrian (Tong et al., 2024) finds that SigLIP (Zhai et al., 2023) is a strong language-supervised model and DINOv2 (Oquab et al., 2023) performs well on vision-centric tasks.

### 2.2 VIDEO LARGE LANGUAGE MODELS

Recent advancements in MLLMs have broadened their application to video understanding tasks. Video LMMs process videos by extracting and encoding frames, then rearranging these as final video features. Several works (Li et al., 2023c; 2024b; Cheng et al., 2024), use the Q-Former module from BLIP-2 to merge visual and text features, while others (Lin et al., 2023; Luo et al., 2023; Ataallah et al., 2024a) concatenate frame features directly.

When processing lengthy videos, the constraint on context length inevitably causes a trade-off between the number of tokens per frame and the number of frames to input. Most existing works (Li et al., 2023c; Ataallah et al., 2024a; Cheng et al., 2024; Zhang et al., 2024b; Li et al., 2024a) address this challenge by uniformly sampling frames from the video, which, however, results in a significant loss of visual details within the video. Video-ChatGPT (Maaz et al., 2023b) employs pooling modules to reduce data dimensions, enhancing processing efficiency. Other works try to preserve the maximum number of frames in video content. LLaMA-VID (Li et al., 2023d) employs an additional text decoder to embed the text query for cross-attention between frame features and compress the context token to one token per frame, while MovieChat (Song et al., 2023) and TimeChat (Ren et al., 2023b) develop memory modules and timestamp-aware encoders to capture detailed video content. Golfish (Ataallah et al., 2024b) segments long videos into shorter clips, processes each segment independently, and retrieves the most relevant segment in response to user queries. MA-LMM (He et al., 2024) maintains a memory bank to aggregate long-term video without exceeding LLMs' context length constraints. LongVILA (Xue et al., 2024) extends the number of video frames to 2048 by enabling 2M context length training. Our work focuses on maximizing the preservation of frames in video content (1fps) within limited context length by proposing spatiotemporal compression of video tokens.

### 2.3 VIDEO TOKEN COMPRESSION

Recent methods has explored dynamic image tokens (Ma et al., 2023; Xu et al., 2022; Bolya et al., 2022) or video tokens (Lee et al., 2024; Ren et al., 2023a; Choi et al., 2024) within the Transformer (Vaswani, 2017) framework. LGDN (Lu et al., 2022) dynamically select salient frames by language-guided supervision for precisely video-language modeling. Chat-UniVi (Jin et al., 2023) extends the dynamic tokens for visual features in MLLMs by merging K-nearest neighbor tokens across frame features of the video input. SlowFast-LLaVA (Xu et al., 2024) uniformly samples 8 frames for high-resolution tokens, while performing spatial pooling to decrease the number of tokens

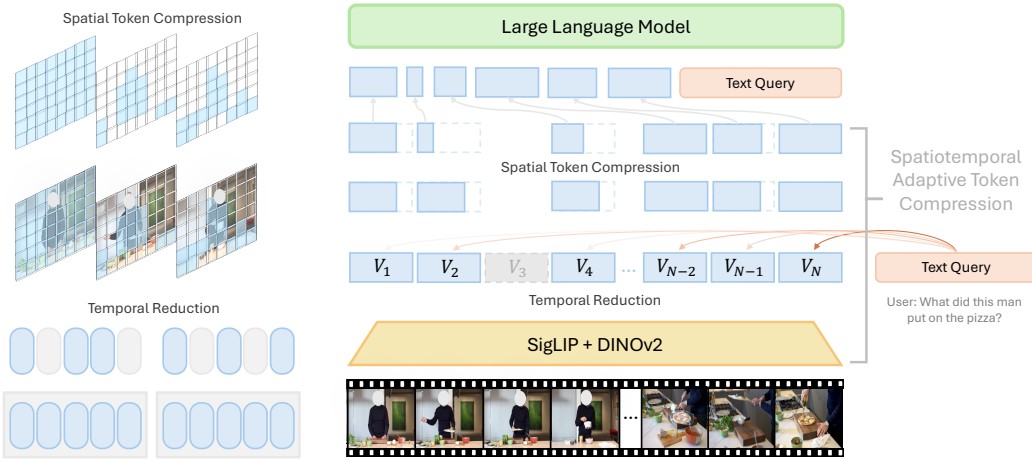

Figure 2: Architecture of LongVU. Given a densely sampled video frames, we first utilize DI-NOv2 (Oquab et al., 2023) prior to remove redundant frames, and fuse the remaining frame features from both SigLIP (Zhai et al., 2023) and DINOv2 (Oquab et al., 2023), described in Section 3.1. Then we selectively reduce visual tokens via cross-modal query, detailed in Section 3.2. Finally, as demonstrated in Section 3.3, we conduct spatial token compression based on temporal dependencies to further meet the limited context length of LLMs.

in frames sampled at a higher frame rate. In our work, we propose a spatiotemporal adaptive token reduction strategy that leverages both cross-modal query and inter-frame dependencies. This approach effectively mitigates temporal redundancy in video content, thereby enabling the accommodation of long videos within a limited context length.

# 3 METHOD

We propose spatiotemporal adaptive compression in three steps to effectively process long video, as shown in Figure 2. Initially, we implement a temporal reduction strategy on the frame sequence by leveraging the prior knowledge from DINOv2 (Oquab et al., 2023) (Section 3.1). Then, we selectively preserve full tokens for key frames via cross-modal query, while applying spatial pooling to reduce the remaining frames into low-resolution token representations (Section 3.2). Furthermore, we implement a spatial token reduction mechanism based on inter-frame temporal dependencies (Section 3.3).

## 3.1 FRAME FEATURE EXTRACTOR AND TEMPORAL REDUCTION

DINOv2 (Oquab et al., 2023), through its self-supervised (SSL) training with a feature similarity objective on vision-centric tasks, can effectively capture subtle frame differences and low-level visual features. In contrast, CLIP-based (Zhai et al., 2023; Radford et al., 2021) models are trained with vision-language contrastive loss in the semantic space, excelling at language alignment while sacrificing low-level features as shown in Figure 6. Moreover, Cambrian (Tong et al., 2024) discovered that combining features from both SigLIP (Zhai et al., 2023) and DINOv2 (Oquab et al., 2023) leads to a significant performance boost in vision-centric tasks. Therefore, we pioneer to leverage both SSL-based model DINOv2 (Oquab et al., 2023) with vision-language contrastive-based model SigLIP (Zhai et al., 2023) as frame feature extractors for MLLM in video understanding task.

Note that processing the entire long video can be computationally expensive. Given a 1fps-sampled video with $N$ frames, denoted as $I = \{I^1, ..., I^N\}$, we first use DINOv2 (Oquab et al., 2023) to extract features from each frame, leading to a set of DINO features $\{V_{\text{dino}}^1, \ldots, V_{\text{dino}}^N\}$. We then calculate the average similarity $\text{sim}^i = \frac{1}{J-1} \sum_{j=1, j \neq i}^{J} \text{sim}(V_{\text{dino}}^i, V_{\text{dino}}^j)$ within each non-overlapping window with $J = 8$ frames and reduce frames that exhibit high similarity with other frames. This

step significantly reduces video redundancy by temporally compressing the original $N$ frames to $T$ frames, which reduces approximately half of the video frames, as detailed in Section 4.6.

We then extract features of the remaining $T$ frames using SigLIP (Zhai et al., 2023) vision encoder, resulting in $T$ features $\{V_{sig}^1, ..., V_{sig}^T\}$. Subsequently, following Cambrian (Tong et al., 2024), we combine these two types of visual features via Spatial Vision Aggregator (SVA) (Tong et al., 2024) that employs learnable queries to spatially aggregate visual features from multiple vision encoders. We denote the fused frames features as $V = \{V^1, ..., V^T\}$.

## 3.2 Selective Feature Reduction via Cross-modal Query

After temporal reduction, we obtain a set of fused frame features from both vision encoders, denoted as $V = \{V^1, ..., V^T\} \in \mathbb{R}^{T \times (H_h \times W_h) \times D_v}$, where $H_h \times W_h$ denotes the spatial dimension of the frame features, and $D_v$ indicates the channel dimension of the frame feature after SVA. If the concatenated frame features exceed the limited context length, i.e., $T \times H_h \times W_h \geq L_{max}$, we develop a selective compression strategy for certain frames, in order to capture both the detailed spatial semantic and long-range temporal context.

To achieve this, we propose using text query to help reduce spatial tokens of certain frames from $H_h \times W_h$ to $H_l \times W_l$. Given the LLM embedding of the text query $Q \in \mathbb{R}^{L_q \times D_q}$, where $L_q$ is the length of text query and $D_q$ is the dimensionality of LLM's embedding space, we strategically choose $N_h$ frames to preserve their original token resolution, while the remaining undergoes a process of spatial pooling to achieve a reduced resolution. The selection mechanism is based on the cross-modal attention scores between each frame feature and the text query. The number of frames to keep original resolution can be formulated as,

$$\mathbf{Top}_{N_h}\left(\frac{1}{H_h W_h L_q} \sum_{h,w,l} \mathcal{F}(V)Q^T\right), \quad N_h = \max\left(0, \frac{L_{\max} - L_q - TH_l W_l}{H_h W_h - H_l W_l}\right), \quad (1)$$

where $L_{max}$ is the maximum context length, $\mathcal{F}(\cdot)$ denotes a multi-layer perceptron (MLP)-based multimodal adapter designed to align visual features with the input space of the LLM. Note that we omit the system prompt in the instruction template for Equation 1 simplification. If $N_h = 0$, indicating that no frames are selected for retention at their original resolution, we will skip the computation of attention scores and will directly perform spatial pooling across all the frames to the lower resolution.

## 3.3 Spatial Token Compression

As previously discussed, there are cases where the concatenated visual features with low resolution tokens still exceeds the maximum context length, i.e., $T \times H_l \times W_l \geq L_{max}$. Under these circumstances, further token compression is necessary. We partition the sequence of frame features into non-overlapping segments with a sliding window of size $K < T$, within which we conduct spatial token compression (STC). The first frame in each window retains its full token resolution. We then compute the cosine similarity between the first frame and subsequent frames within the window, conducting an element-wise comparison of spatial tokens between the first frame and its successors. Spatial tokens that exhibit a cosine similarity $\text{sim}(\cdot, \cdot)$ greater than the threshold $\theta$ with the corresponding tokens of the first frame at the same spatial location will be pruned, which can be formulated as,

$$v_i^* \leftarrow \begin{cases} v_i(h, w) & \text{sim}(v_1(h, w), v_i(h, w)) \leq \theta \\ \emptyset & \text{otherwise} \end{cases}, \quad \forall h \in [1, H_l], w \in [1, W_l], i \in [2, K] \quad (2)$$

Given that videos often contain significant pixel-level redundancy, particularly in static background, this method allows spatial tokens reduction via temporal dependencies. We chose the first frame in each sliding window for comparison, assuming DINOv2 (Oquab et al., 2023) has effectively reduced video redundancy across frames, making each frame less similar. We also tested alternative strategies, like using the middle frame or adaptively selecting based on frame changes (Section 4.5), but these provided similar performance and compression rates. Therefore, we chose the first-frame strategy in each sliding window for its simplicity and effectiveness.

# 4 EXPERIMENTS

## 4.1 DATASETS

We adopt two stages of training in our experiments: image-language pre-training and video-language finetuning. For the image-language pre-training stage, previous methods (Chen et al., 2023b; Peng et al., 2023; Wang et al., 2023; Chen et al., 2023a; Liu et al., 2024b; Dong et al., 2024) usually use two steps for alignment and finetuning. For simplicity, we combine these two steps in one stage using Single-Image data from LLaVA-OneVision (Li et al., 2024a). For video-language finetuning, we utilize a large-scale video-text pairs sourced from several publicly accessible databases. The video training data contains a subset of VideoChat2-IT (Li et al., 2024b), which includes TextVR (Wu et al., 2025), Youcook2 (Zhou et al., 2018), Kinetics-710 (Kay et al., 2017), NExTQA (Xiao et al., 2021), CLEVRER (Yi et al., 2019), EgoQA (Fan, 2019), TGIF (Li et al., 2016), WebVidQA (Yang et al., 2021), ShareGPT4Video (Chen et al., 2024), and MovieChat (Song et al., 2024) as the long video complementary. All the training datasets are listed in Table 6.

## 4.2 BENCHMARKS AND METRICS

We evaluate our model on EgoSchema (Mangalam et al., 2024), MVBench (Li et al., 2024b), VideoMME (Fu et al., 2024) and MLVU (Zhou et al., 2024). VideoMME (Fu et al., 2024) (1 min $\sim$ 1 hour) and MLVU (Zhou et al., 2024) (3 mins $\sim$ 2 hours) are long video benchmarks for assessing long video understanding ability. For VideoMME (Fu et al., 2024), videos are officially split based on duration, which contains a subset of long videos ranging from 30 minutes to 1 hour. We perform standardized evaluations using greedy decoding (*num_beams*=1) and benchmark our results against other open-source and proprietary models.

| Models | Size | Context Length | #Frames | EgoSchema | MVBench | MLVU | VideoMME Overall | VideoMME Long |
|---|---|---|---|---|---|---|---|---|
| Duration | | | | 179.8 sec | 16 sec | 3∼120 min | 1∼60 min | 30∼60 min |
| *Proprietary Models* | | | | | | | | |
| GPT4-V (OpenAI, 2023) | - | - | 1fps | 55.6 | 43.7 | - | 60.7 | 56.9 |
| GPT4-o (OpenAI, 2024) | - | - | 1fps | 72.2 | - | 64.6 | 77.2 | 72.1 |
| *Open-Source Video MLLMs* | | | | | | | | |
| Video-LLaVA (Lin et al., 2023) | 7B | 4k | 8 | 38.4 | 41.0 | 47.3 | 40.4 | 38.1 |
| LLaMA-VID (Li et al., 2023d) | 7B | 4k | 1fps | 38.5 | 41.9 | 33.2 | - | - |
| Chat-UniVi (Jin et al., 2023) | 7B | 4k | 64 | - | - | - | 45.9 | 41.8 |
| ShareGPT4Video (Chen et al., 2024) | 8B | 8k | 16 | - | 51.2 | 46.4 | 43.6 | 37.9 |
| LLaVA-NeXT-Video (Zhang et al., 2024b) | 7B | 8k | 32 | 43.9 | 33.7 | - | 46.5 | - |
| VideoLLaMA2 (Cheng et al., 2024) | 7B | 8k | 32 | 51.7 | 54.6 | 48.5 | 46.6 | 43.8 |
| LongVA (Zhang et al., 2024a) | 7B | 224k | 128 | - | - | 56.3 | 54.3 | 47.6 |
| VideoChat2 (Li et al., 2024b) | 7B | 8k | 16 | 54.4 | 60.4 | 47.9 | 54.6 | 39.2 |
| LLaVA-OneVision (Li et al., 2024a) | 7B | 8k | 32 | 60.1 | 56.7 | 64.7 | 58.2 | 46.7 |
| LongVU (Ours) | 7B | 8k | 1fps | **67.6** | **66.9** | **65.4** | **60.6** | **59.5** |

Table 1: Results on comprehensive video understanding benchmarks

## 4.3 IMPLEMENTATION DETAILS

We use SigLIP (Zhai et al., 2023) (so400m-patch14-384) and DINOv2 (Oquab et al., 2023) as the vision encoder while choose Qwen2-7B (Qwen, 2024) and Llama3.2-3B (Llama, 2024) as our language foundation model. We only compute cross-entropy loss for autoregressive text generation. We use AdamW (Loshchilov, 2017) optimizer with a cosine schedule for all the trainings. In the image-language pre-training stage, we train the model for one epoch with global batch size of 128. The learning rate is set to 1e-5, and the warmup rate is 0.03. The number of tokens per image are set to 576. For the video-language finetuning stage, we train the model for one epoch with global batch size of 64. The learning rate is set to 1e-5, and the warmup rate is 0.03. The maximum number of tokens per frame are set to 144 ($H_h = W_h = 12$), while each might be reduced by our proposed adaptive compression approach ($\leq 64$, $H_l = W_l = 8$). The DINO threshold is set as 0.83 and the STC reduction threshold is $\theta = 0.75$. The sliding window size $K = 8$. Our model is trained on 64 NVIDIA H100 GPUs.

## 4.4 VIDEO UNDERSTANDING

**Quantitative Results.** Table 1 presents our experimental results on multiple video understanding benchmarks. Our results compares favorably to all the baselines across various video understanding benchmarks. For example, on VideoMME (Fu et al., 2024), our LongVU outperforms VideoChat2 (Li et al., 2024b), LLaVA-OneVision (Li et al., 2024a) by 6.0% and 2.4% respectively. Notably, on VideoMME Long subset (Fu et al., 2024), our model surpasses LLaVA-OneVision (Li et al., 2024a) by 12.8%. These results indicate the strong video understanding capabilities of our model. Note that our model achieves significant improved performance with a much smaller training dataset, comparing to LLaVA-OneVision (Li et al., 2024a) trained on OneVision-1.6M (multi-image, video) that has not yet been made publicly available[1]. With the same video training dataset from VideoChat2-IT (Li et al., 2024b), our LongVU shows much higher performance than VideoChat2 (Li et al., 2024b), ∼10% accuracy improvement in average. Interestingly, we also find that our model can even beat proprietary model GPT4-V (OpenAI, 2023) on MVBench (Li et al., 2024b) with densely sampled video input and reduce the accuracy gap comparing to proprietary models on other video benchmarks.

We also scale our LongVU with a lightweight LLM, Llama3.2-3B (Llama, 2024), to further demonstrate the strong video understanding capabilities. We observe the consistent improvement of our light-weight LongVU over baselines in Table 2. Our method outperforms Phi-3.5-vision-instruct (Abdin et al., 2024) on VideoMME (Long) by margin of 3.4% accuracy. This set of experiments validate the effectiveness of our method even scaling to a smaller size.

| Models | EgoSchema | MVBench | VideoMME | | MLVU |
| --- | --- | --- | --- | --- | --- |
| | | | Overall | Long | |
| InternVL2 (InternLM2-1.8B) (OpenGVLab, 2024) | - | 60.2 | 47.3 | 42.6 | - |
| VideoChat2 (Phi-3-mini-4B) (Li et al., 2024b) | 56.7 | 55.1 | - | - | - |
| Phi-3.5-vision-instruct (Phi-3-mini-4B) (Abdin et al., 2024) | - | - | 50.8 | 43.8 | - |
| LongVU (Ours) (Llama3.2-3B) | **59.1** | **60.9** | **51.5** | **47.2** | 55.9 |

Table 2: Results of small-size video language models across video understanding benchmarks.

**Qualitative Results.** We now provide the qualitative results in Figure 3. Specifically, we demonstrate various video understanding abilities in the examples, such as accurately recognizing the orientation of moving objects in Figure 3(a), providing detailed video descriptions in Figure 3(b), identifying inserted needle frames and conducting action counting in Figure 3(c), and responding precisely to questions about specific frames in an hour-long video in Figure 3(d). These results demonstrate that our model has competing video-language understanding capabilities.

## 4.5 ABLATION STUDIES

**Effects of the number of tokens per frame.** We ablate the number of tokens in our uniform-sampling baselines. There is a trade-off between the number of tokens per frame and the sampling frequency of frames. Table 3 shows the experimental results when using different number of tokens with different sampling. When applying uniforming sampling, 144 tokens per frame shows better performance than 64 tokens in an 8k context length on VideoMME (Fu et al., 2024) and MLVU Zhou et al. (2024) while worse on EgoSchema Mangalam et al. (2024). With 144 tokens per frame, it preserves more visual details, but restricts the total number of frames, i.e., less than 60 frames within 8k context length. This demonstrate that adaptive tokens are needed for better performance across different video benchmarks.

**DINOv2 vs SigLIP.** Our results in Table 3 verify that DINOv2 (Oquab et al., 2023) features are more effective than SigLIP (Zhai et al., 2023) features. As expected, we also find that using DINO-based features for temporal frame reduction outperforms uniform sampling. Therefore, DINOv2 (Oquab et al., 2023) is an useful vision-centric feature extractor to help perform temporal reduction.

**Query guided selection.** We apply text-guided frame selection after temporal reduction, where relevant frames are maintained at full token capacity (144 tokens), while others are reduced to 64

---

[1]LLaVA-OneVision (Li et al., 2024a) only release single-image set at the time of current submission. https://huggingface.co/datasets/lmms-lab/LLaVA-OneVision-Data/discussions/6

00:00:00 ———————————————————————→ 00:00:05

> Can you identify the direction of the yellow sphere's movement in the video?

> The yellow sphere is moving away from the red cube and towards the right side of the frame.

(a) Spatial-temporal Orientation Awareness

00:00:00 ———————————————————————→ 00:00:20

> Describe the video in detail.

> The video begins with a scene featuring two animated characters in a fantastical setting, suggesting a narrative of adventure or conflict. The first character, dressed in a yellow and red martial arts uniform with a mask, is seen in a defensive or preparatory stance, while the second character, an older man with a white beard and a blue robe, appears surprised or concerned. The background is filled with green, leaf-like structures and a mountainous landscape, indicating a natural, possibly magical environment.

(b) Video Detailed Description

00:00:00 ———————————————————————→ 00:10:53

> In this video, how many times does the scene of the 'cleaning toilet' action appear in total?

> Two.

(c) Action Counting

00:00:00 ———————————————————————→ 01:40:08

> What type of view is provided of the tropical beach in the video?

> The video provides an aerial view of the tropical beach.

(d) Hour-long Video Understanding

Figure 3: Examples for various video understanding capabilities of LongVU model. We showcase that our LongVU is able to completing different types of video understanding tasks.

tokens. This helps preserve essential visual features and accommodates more long-range context within the context length. In Table 3, we observe the improvement with query guided frame selection across all benchmarks. Moreover, in Table 4, the results of each subtask in MLVU (Zhou et al., 2024) show significant performance improvements when using cross-modal queries, particularly for frame-retrieval tasks such as counting and needle detection.

**Spatial token compression.** We further apply spatial token compression after query guided selection. We find that spatial token compression (STC) not only enhances performance within 8k context length, but also achieve results comparable or slightly better than 16k context length in Table 3. We also note some improvements for most subtasks in MLVU (Zhou et al., 2024).

| Methods | Context Length | #Tokens | EgoSchema | VideoMME | MLVU |
|---|---|---|---|---|---|
| Uniform | 16k | 144 | 67.12 | 60.01 | 64.70 |
| DINO | 16k | 144 | 67.34 | 61.25 | 64.83 |
| Uniform | 8k | 64 | 66.84 | 57.56 | 60.87 |
| Uniform | 8k | 144 | 66.28 | 58.84 | 63.28 |
| SigLIP | 8k | 64 | 66.04 | 58.63 | 62.17 |
| DINO | 8k | 64 | 66.20 | 59.90 | 62.54 |
| DINO + Query | 8k | 64/144 | 67.30 | 60.08 | 65.05 |
| DINO + Query + STC (default) | 8k | dynamic | **67.62** | **60.56** | **65.44** |

Table 3: Ablation studies of number of tokens per frame, different context lengths, and our spatiotemporal compression components.

| Stratgy | count | ego | needle | order | plotQA | anomaly | reasoning | Avg |
|---|---|---|---|---|---|---|---|---|
| DINO | 24.15 | 59.09 | 68.16 | 52.89 | 71.24 | 74.00 | 86.36 | 62.54 |
| DINO+Query | 28.98 | 55.39 | **78.87** | 56.37 | **72.35** | 75.50 | **87.87** | 65.05 |
| DINO+Query+STC (default) | **28.98** | **59.37** | 76.33 | **58.30** | 71.61 | **76.00** | 87.50 | **65.44** |

Table 4: Ablation study on each subtask in MLVU (Zhou et al., 2024).

**Different strategies for spatial token compression.** We now ablate different strategies of our spatial token compression mechanism. This analysis explores different strategies for determining anchor frames: the first/middle one in each sliding window, or the frame that exhibits significant changes compared to its adjacent frames. In Table 5, our results indicate that taking the first frame in each sliding window gives a slightly better performance with similar reduction rates across all strategies.

| Model | Short | Medium | Long | Overall | Reduction rate |
|---|---|---|---|---|---|
| $1^{st}$ frame in sliding window (default) | 64.7 | 58.2 | 59.5 | 60.9 | 55.47% |
| $(K/2)^{th}$ frame in sliding window | 64.7 | 58.7 | 58.6 | 60.7 | 54.97% |
| frame with high changes | 64.7 | 58.2 | 58.3 | 60.4 | 55.62% |

Table 5: Different strategies for spatial token compression on VideoMME (Fu et al., 2024).

## 4.6 SPATIOTEMPORAL COMPRESSION ANALYSIS

**Compression analysis.** We sampled hundreds of videos to demonstrate the distribution of frame/token reduction rate. Figure 4 (a) presents the number of frames before and after temporal reduction based on the similarity of DINOv2 features across frames. We find that ~45.9% of the frames are maintained after temporal reduction on average. Figure 4 (b) shows the number of tokens before and after spatial token compression (Section 3.3). We observe that ~40.4% tokens are reduced on average. These results demonstrate the effective video token compression with temporal and spatial token reduction.

**Long context analysis.** Recently, the Needle-in-a-Haystack task (Hsieh et al., 2024; Kamradt., 2023) has been used to assess the ability of Large Language Models (LLMs) to retrieve long context

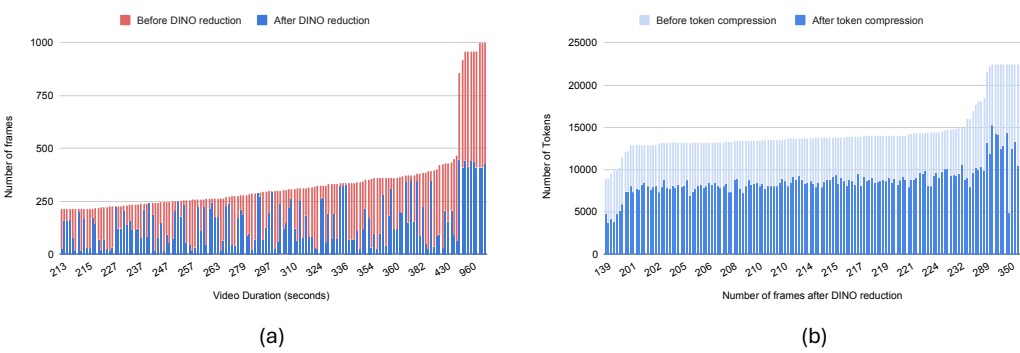

(a)                                                                 (b)

Figure 4: We randomly sample hundreds of videos to demonstrate the frames/tokens level reduction rate. (a) The number of frames before/after temporal reduction based on DINOv2 features (Section 3.1). (b) The number of tokens before/after spatial token compression (Section 3.3).

information. We follow (Zhang et al., 2024a) to conduct a video needle-in-a-haystack experiment to demonstrate the effectiveness of our compression strategy on identifying the needle frame within an hour-long video.

To facilitate this evaluation, we randomly select an one-hour-long test video from MLVU (Zhou et al., 2024). We then insert each image from a set of VQA problems as a needle frame into this long video for creating a challenging search task. We sample the video at 1 FPS and control the frame length ranging from 200 to 3.6k frames. We also vary the needle frame insertion depth from 0% to 100% of the total input frames. We conduct experiments with 8k context length and compare our adaptive token compression to the one without applying query-guided selection (w/o Query) and spatial token compression (w/o STC) after temporal reduction. Figure 5 demonstrates that our adaptive compression mechanism could accurately resolve the needle VQA problem of 1k frames within 8k context length and improve score with more frames. This demonstrates the advantage of our method for long context video understanding.

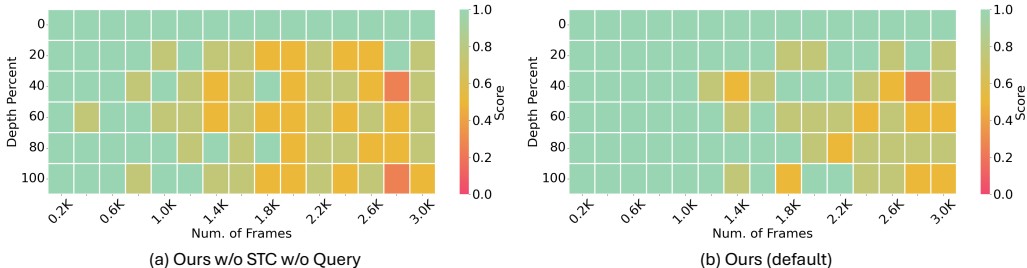

(a) Ours w/o STC w/o Query                                   (b) Ours (default)

Figure 5: Needle-in-a-Haystack results. Our adaptive token compression scheme improves the score for locating the needle frame within an hour-long video from 0.80 to 0.88 on average.

## 5 CONCLUSION

We introduced LongVU, a MLLM that can address the significant challenge of long video understanding within a limited context length. To achieve this, we proposed a spatiotemporal adaptive compression scheme of LongVU for helping reduce video tokens without losing much visual details of long videos by leveraging cross-modal query and inter-frame similarities. Experiments on various video understanding benchmarks consistently validate the advantages of our model. We also demonstrate that our method helps build a quality light-weight video language understanding model based on Llama3.2-3B, which suggests that LongVU has many potential applications in the vision-language community.

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

## A  TRAINING DATASETS

For the image-language training stage, previous methods (Chen et al., 2023b; Peng et al., 2023; Wang et al., 2023; Chen et al., 2023a; Liu et al., 2024b; Dong et al., 2024) usually use two stages of alignment and finetuning. For simplicity, we combine and alignment in one stage using single image version of LLaVA-OneVision (Li et al., 2024a) data. For video-language training, we utilize a large-scale video-text pairs sourced from several publicly accessible databases. The video training data is a subset of VideoChat2-IT (Li et al., 2024b), which includes TextVR (Wu et al., 2025), Youcook2 (Zhou et al., 2018), Kinetics-710 (Kay et al., 2017), NExTQA (Xiao et al., 2021), CLEVRER (Yi et al., 2019), EgoQA (Fan, 2019), TGIF (Li et al., 2016), WebVidQA (Yang et al., 2021), ShareGPT4Video (Chen et al., 2024), in addition to above, we use MovieChat (Song et al., 2024) as long video complementary. All the training data is demonstrated in Table 6.

## B  FRAME-LEVEL POSITION ENCODING

To alleviate potential confusion arising from frame-by-frame feature concatenation, we incorporate a frame-level position encoding to enforce the temporal boundaries across frames and capture interdependencies within each frame. Given that we temporally reduce several frames, a straightforward concatenation of all frames renders the model unaware of the relative timestep across frames. Furthermore, our dynamic token sampling strategy does not delineate clear boundaries between each frame. To address this, we incorporate frame-level positional embeddings (FPE) that correspond to

| Modality | Task | # Samples | Dataset |
|----------|------|-----------|---------|
| Image-Text | Single-Image | 3.2M | LLaVA-OneVision |
| Video-Text | Captioning | 43K | TextVR, MovieChat, YouCook2 |
| | Classification | 1K | Kinetics-710 |
| | VQA | 424K | NExTQA, CLEVRER, EgoQA, TGIF, WebVidQA, DiDeMo |
| | Instruction | 85K | ShareGPT4Video |

Table 6: Training data statistics.

| Model | Size | Frames | Short | Medium | Long | Overall |
|-------|------|--------|-------|--------|------|---------|
| Video-LLaVA (Lin et al., 2023) | 7B | 8 | 46.1 | 40.7 | 38.1 | 41.6 |
| ShareGPT4Video (Chen et al., 2024) | 8B | 16 | 53.6 | 39.3 | 37.9 | 43.6 |
| Chat-Univi-v1.5 (Jin et al., 2023) | 7B | 64 | 51.2 | 44.6 | 41.8 | 45.9 |
| VideoLLaMA2 (Cheng et al., 2024) | 7B | 16 | 59.4 | 47.6 | 43.8 | 50.3 |
| VideoChat2 (Li et al., 2024b) | 7B | 16 | 52.8 | 39.4 | 39.2 | 43.8 |
| LongVA (Zhang et al., 2024a) | 7B | 128 | 61.6 | 50.4 | 47.6 | 54.3 |
| LLaVA-OneVision (Li et al., 2024a) | 7B | 32 | **69.1** | 53.3 | 46.7 | 58.2 |
| LongVU (Ours) | 7B | 1fps | 64.7 | **58.2** | **59.5** | **60.9** |

Table 7: Comparison with other video LMMs on VideoMME (Fu et al., 2024) benchmark.

the absolute timestep of each frame, utilizing a shared sinusoidal position encoding (Vaswani, 2017) for frames at time $t$, shown in Equation 3.

$$PE(t, 2i) = sin(t/10000^{2i/d}), PE(t, 2i + 1) = cos(t/10000^{2i/d}) \tag{3}$$

The ablation shows in Table 8 and Table 9 that adding the FPE does not affect much to the overall performance across several benchmarks. Therefore, we decide not to include it in our default setting.

| Methods | Context Length | #Tokens | EgoSchema | VideoMME | MLVU |
|---------|----------------|---------|-----------|----------|------|
| DINO + Query | 8k | 64/144 | 67.30 | 60.08 | 65.05 |
| DINO + Query + STC (default) | 8k | dynamic | 67.62 | 60.56 | 65.44 |
| DINO + Query + STC + FPE | 8k | dynamic | 67.87 | 60.89 | 64.56 |

Table 8: Ablation study on with or without FPE.

| Stratgy | count | ego | needle | order | plotQA | anomaly | reasoning | Avg |
|---------|-------|-----|--------|-------|--------|---------|-----------|-----|
| DINO | 24.15 | 59.09 | 68.16 | 52.89 | 71.24 | 74.0 | 86.36 | 62.54 |
| DINO+Query | 28.98 | 55.39 | 78.87 | 56.37 | 72.35 | 75.5 | 87.87 | 65.05 |
| DINO+Query+STC (default) | 28.98 | 59.37 | 76.33 | 58.30 | 71.61 | 76.0 | 87.50 | 65.44 |
| DINO+Query+STC+ FPE | 29.46 | 60.79 | 74.08 | 52.12 | 71.79 | 74.5 | 86.74 | 64.56 |

Table 9: Strategy ablations on each subtask in MLVU (Zhou et al., 2024).

## C   DINOV2 V.S. SIGLIP

DINOv2 (Oquab et al., 2023), through self-supervised training with a feature similarity objective on visually-centric tasks, captures subtle frame differences and low-level visual features more effectively

than vision-language contrastive methods (Radford et al., 2021; Zhai et al., 2023), as shown in Figure 6.

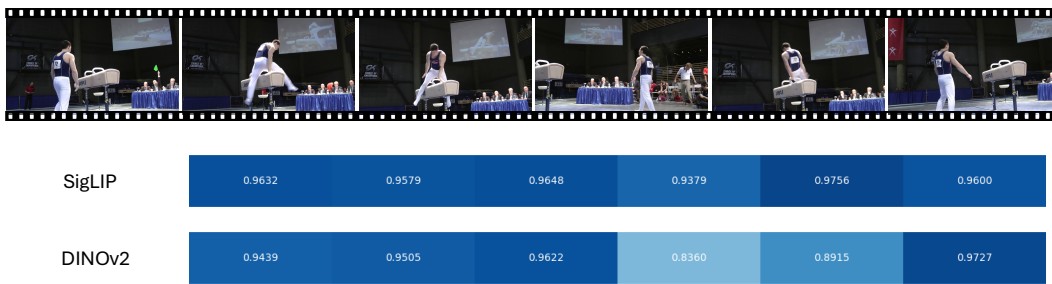

| | | | | | | |
|---|---|---|---|---|---|---|
| SigLIP | 0.9632 | 0.9579 | 0.9648 | 0.9379 | 0.9756 | 0.9600 |
| DINOv2 | 0.9439 | 0.9505 | 0.9622 | 0.8360 | 0.8915 | 0.9727 |

Figure 6: Similarity comparison between SigLIP (Zhai et al., 2023) and DINOv2 (Oquab et al., 2023) features. The similarity is calculated between the first frame and the remainings. DINO concentrating on vision centric task effectively capture subtle frame differences compared with SigLIP (Zhai et al., 2023) which is aligned on semantic space.

## D NEEDLE-IN-A-VIDEO-HAYSTACK

We conducted experiments using an 8k context length to evaluate our default setting, which incorporates our adaptive compression, against configurations without spatial token compression (w/o STC) and without querying guided reduction (w/o Query), as depicted in Figure 7. By integrating a cross-modal query to selectively retain full tokens of frames relevant to the text query, the model significantly enhances its ability to accurately identify key frames when the total number of video frames is fewer than 1.4k. Moreover, our adaptive token compression mechanism further boosts VQA accuracy with increased frames.

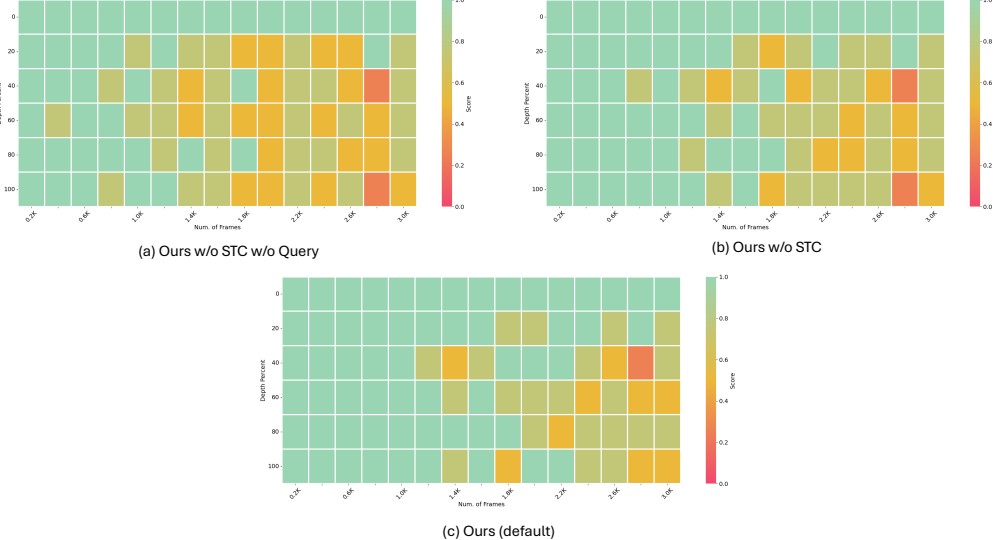

Figure 7: Needle-In-A-Video-Haystack results. Our spatiotemporal adaptive token compression scheme improves the score for locating the needle frame.

## E INFERENCE TIME

To evaluate the computational overhead introduced by our proposed spatiotemporal compression approach, we compare it with various baselines using input videos of the same length (20 minutes)

| Model | SQA-IMG | MMVP | POPE | RealWorldQA |
|---|---|---|---|---|
| Before video SFT | 95.44 | 51.33 | 86.65 | 61.06 |
| After video SFT | 83.94 | 32.00 | 81.23 | 47.65 |

Table 10: We mainly focus on video understanding task and use video-only data for video SFT stage. We observe a decrease in performance on image understanding after video SFT stage.

sampled at 1 FPS. The experiments were conducted on an A100 GPU with 80 GB memory. LLaMA-VID (Li et al., 2023d) encounters a CUDA out-of-memory (OOM) issue when processing 20-minute videos as input. Our method demonstrates faster performance compared to the token compression approach of Chat-UniVi (Jin et al., 2023), which relies on a KNN-based strategy to merge similar tokens. Furthermore, it is more efficient than the resampler-based method VideoChat2 (Li et al., 2023c), which compresses video inputs using learnable queries in Q-Former. When compared to methods without compression, such as LLaVA-OneVision (Li et al., 2024a), our approach is slightly slower, requiring 1.27x the processing time.

| Models | LLaMA-VID | Chat-UniVi | VideoLLaMA2 | VideoChat2 | LLaVA-OneVision | Ours |
|---|---|---|---|---|---|---|
| Time (sec) | OOM | 49.06 | 58.62 | 45.22 | 25.84 | 32.96 |

Table 11: Inference time comparison on a 20 minutes videos. All models take frames sampled at 1fps as input, approximately 1200 frames.

We begin by using the DINOv2 vision encoder to extract features from all frames and then reduce redundant frames based on DINO feature similarity. After this reduction, the remaining frames are processed using SigLIP. One significant advantage of our method is that the DINO-based frame reduction step substantially decreases the computation required for the remaining frames in subsequent steps. As shown in the table below, the primary computation lies in frame feature extraction, which, in real-world applications, can be preprocessed offline. Notably, our proposed compression component contributes only a small portion to the overall inference overhead.

| Component | Extract DINO feature | DINO similarity | Extract SigLIP feature | Query | STC |
|---|---|---|---|---|---|
| Time (sec) | 22.2 | 1.05 | 4.32 | 0.27 | 0.18 |

Table 12: Inference time of each component.

## F ABLATIONS

| Context length | EgoSchema | MVBench | MLVU | VideoMME |
|---|---|---|---|---|
| 6k | 67.82 | 66.71 | 62.33 | 59.54 |
| 8k (default) | 67.6 | 66.9 | 65.4 | 60.6 |
| 12k | 67.14 | 66.83 | 63.54 | 60.12 |
| 16k | 67.20 | 66.86 | 64.4 | 60.2 |

Table 13: Context length ablation.

## G LIMITATION

Our research is primarily concentrated on video understanding tasks, for which we employ video-only data during the video supervised fine-tuning (SFT) stage. As evidenced in Table 10, there is a decrease

| DINO threshold | EgoSchema | MVBench | MLVU | VideoMME |
|---|---|---|---|---|
| 0.9 | 67.64 | 66.88 | 64.33 | 60.3 |
| 0.85 | 67.66 | 66.86 | 63.12 | 59.9 |
| 0.83 (default) | 67.6 | 66.9 | 65.4 | 60.6 |
| 0.8 | 67.18 | 66.86 | 63.51 | 60.34 |
| 0.75 | 67.22 | 66.86 | 63.16 | 60.38 |

Table 14: DINO threshold ablation.

| STC threshold | EgoSchema | MVBench | MLVU | VideoMME |
|---|---|---|---|---|
| 0.85 | 67.56 | 66.88 | 64 | 59.98 |
| 0.8 | 67.3 | 66.86 | 63.51 | 59.83 |
| 0.75 (default) | 67.6 | 66.9 | 65.4 | 60.6 |
| 0.7 | 67.5 | 66.86 | 64.03 | 60.27 |
| 0.65 | 67.42 | 66.86 | 63.91 | 60.34 |

Table 15: STC threshold ablation.

| Sliding window $K$ | EgoSchema | MVBench | MLVU | VideoMME |
|---|---|---|---|---|
| 4 | 67.38 | 66.86 | 63.74 | 60.45 |
| 8 (default) | 67.6 | 66.9 | 65.4 | 60.6 |
| 16 | 67.22 | 66.86 | 62.18 | 60.42 |
| 32 | 67.2 | 66.86 | 60.69 | 60.82 |

Table 16: Sliding window $K$ ablation.

| Sliding window $J$ | EgoSchema | MVBench | MLVU | VideoMME |
|---|---|---|---|---|
| 4 | 67.54 | 66.88 | 63.79 | 60.6 |
| 8 (default) | 67.6 | 66.9 | 65.4 | 60.6 |
| 16 | 67.56 | 66.86 | 64.3 | 60.16 |
| 32 | 67.54 | 66.83 | 63.38 | 60.23 |

Table 17: Sliding window $J$ ablation.

observed in the model's image understanding capabilities after video SFT. A potential remedy could involve integrating a mix of image, multi-image, and video data during training. However, due to constraints in GPU resources, we leave it as a future work with larger datasets for stronger unified image and video models.

Our method spatiotemporally reduces video frames/tokens and concatenates tokens all together to form the overall video representation. However, this approach does not encode the temporal location of individual frames. While we experimented with frame-level positional embeddings to alleviate this drawback, the model still struggles with tasks like temporal grounding, meaningly identifying the precise start and end times of events.

We think that a well-designed frame-level positional embedding could help address this issue. Alternatively, explicitly adding <frame_i> text to demonstrate the timestamp of each frame or overlaying visual text on the frames to indicate their timestamps could also be a potential solution.

