# OpenReview forum: "LongVU: Spatiotemporal Adaptive Compression for Long Video-Language Understanding"
_ICLR.cc/2025/Conference — Submitted to ICLR 2025_

### Official Review · Reviewer_kRvH · 2024-10-25

**Soundness:** 3
**Presentation:** 3
**Contribution:** 2
**Rating:** 5
**Confidence:** 5

**Summary:**

The paper proposes a video compression strategy, including: removing redundant frames, selective frame feature reduction, and spatial token reduction. This strategy effectively reduces the number of visual tokens in MLLM and significantly improves MLLM performance on long videos.

**Strengths:**

- The paper is generally well-written and easy to understand.
- The performance is encouraging.

**Weaknesses:**

- There are many details in the pipeline that are not explained clearly.

1. Line 52: The statement "the resampling module causes important visual information loss" needs further explanation. The resampling module is widely used in MLLMs, such as in the latest **qwen2-vl** and **MiniCPMV-2.6**.

2. Equation 1 lacks a detailed explanation, and there are errors in the symbol definitions. For example, in line 242, why is it not $L_{max}$?

3. The setup in Section 3.1 lacks necessary details, such as why $ J $ is set to 8.

4. In line 215, the statement "about half of the video frames were reduced" depends on the hyperparameter settings, but the paper does not explain how the hyperparameters were chosen.

5. Figure 4(a) shows the comparison of the number of frames before and after compression. However, this result depends on the choice of threshold, which is not explained in the paper.

6. In Table 4, the **STC** strategy is not always effective. What is the reason for this?

7. Line 207: The authors claim, "we pioneer to leverage both SSL-based model DINOv2 with vision-language contrastive-based model SigLIP as frame feature extractors for MLLM in video understanding tasks." In fact, several similar works already exist [1-2].

[1] Eyes Wide Shut? Exploring the Visual Shortcomings of Multimodal LLMs

[2] Cambrian-1: A Fully Open, Vision-Centric Exploration of Multimodal LLMs

- Compared to previous MLLMs, the three proposed modules in the paper require additional computational processes, such as inter-frame similarity calculation and frame-question similarity calculation. These dense computations will undoubtedly significantly increase inference time. However, the paper lacks a discussion on inference time.

- The proposed three compression strategies seem to be transferable to existing MLLMs in a training-free manner.

**Questions:**

See weakness.

---

> ### Author Response · Authors · 2024-11-22
> **Response to Reviewer kRvH Part [1/2]**
>
> We sincerely thank you for your detailed review. We would like to provide clarifications to all the points you raised by responding to each of your questions below.
>
> **W1: Detail explanation**
>
> **(1) Line 52: The statement "the resampling module causes important visual information loss" needs further explanation. The resampling module is widely used in MLLMs, such as in the latest Qwen2-VL and MiniCPMV-2.6.**
>
> Thanks for sharing those newly released works. We will include this in our related work discussion. However, both of the mentioned papers did not use a resampling module for video understanding. MiniCPM-V [1] (technical report for MiniCPM-V 2.5) conducts image partition to divide image into multiple slices, each slice will be encoded into 1024 tokens. They use 64 learnable queries to pass through a one-layer cross-attention module to compress the tokens for each slice. Therefore, they propose a learnable query to compress the image, and there is no video resampling module. The reviewer mentioned MiniCPM-V 2.6 (a MLLM supports video input) did not provide any technical report, and by checking the repo, we assume it follows the same model architecture as MiniCPM-V 2.5, which only has an image resampler.
>
> Qwen2-VL [2] modifies the ViT for dynamic resolution input, and use a simple MLP layer to compress adjacent 2 × 2 tokens into a single token, which does not contain any resampling module. In addition, Qwen2-VL is also token consuming, for example, it uses 2208 tokens to represent a 16-second video resolution 336x644.
>
> This comment might distort our original meaning. In line 52 our original expression is that some approaches "employ **intensive resampling** modules leading to a considerable loss of essential visual information".
>
> [1] MiniCPM-V: A GPT-4V Level MLLM on Your Phone
>
> [2] Qwen2-VL: Enhancing Vision-Language Model's Perception of the World at Any Resolution
>
> **(2) Why not L_max in line 242?**
>
> Thanks for pointing this out. There is a typo in line 242, where $L_q$ should be $L_{max}$.  We have fixed this notation in our revised PDF.
>
> **(3) Why is J set to 8?**
>
> $J$ represents the size of the sliding window, a minor hyperparameter used to divide the video into chunks. Below, we present ablation results demonstrating the effects of different values for $J$. Adjusting the sliding window size has minimal impact on the benchmarks.
>
> | Sliding window $J$ | EgoSchema | MVBench | MLVU | VideoMME |
> | -------- | ------- | ------- | ------- | ------- |
> | 4 | 67.54 | 66.88 | 63.79 | 60.60 |
> | 8 (default) | 67.6 | 66.9 | 65.4 | 60.6 |
> | 16 | 67.56 | 66.86 | 64.3 | 60.16 |
> | 32 | 67.54 | 66.83 | 63.38 | 60.23 |
>
> **(4)(5) Hyperparameters chosen for frame reduction**
>
> We empirically observe DINO similarities with several videos and set the DINO threshold as 0.83 for all the benchmarks. We also ablate the DINO threshold. The lower the DINO threshold, the fewer frames are reduced. For instance, in MLVU, as the threshold decreases, performance deteriorates due to the retention of more redundant frames within a given context length.
>
> | DINO threshold | EgoSchema | MVBench | MLVU | VideoMME |
> | -------- | ------- | ------- | ------- | ------- |
> | 0.9 | 67.64 | 66.88 | 64.33 | 60.30 |
> | 0.85 | 67.66 | 66.86 | 63.12 | 59.90 |
> | 0.83 (default) | 67.6 | 66.9 | 65.4 | 60.6 |
> | 0.8 | 67.18 | 66.86 | 63.51 | 60.34 |
> | 0.75 | 67.22 | 66.86 | 63.16 | 60.38 |
>
> **(6) In Table 4, the STC strategy is not always effective. What is the reason for this?**
>
> Table 4 presents the ablation results for each subtask of MLVU. Although there is a slight performance drop in 3 out of the 7 subtasks, this can be attributed to the compression potentially affecting questions that might rely heavily on fine visual details. However, our default setting (DINO+Query+STC) achieves an average overall improvement of 0.39% compared with DINO+Query, delivering the best results.

---

> ### Author Response · Authors · 2024-11-22
> **Response to Reviewer kRvH Part [2/2]**
>
> **(7) The authors claim, "we pioneer to leverage both DINOv2 and SigLIP in video understanding tasks." In fact, several similar works already exist.**
>
> This comment may misrepresent our original intent. In this sentence (Line 207), we want to emphasize that we utilize both DINO and SigLIP for MLLM in in **video understanding tasks**. To the best of our knowledge, Eyes Wide Shut [1] and Cambrian [2] focus only on image modality and haven't explored the benefits of DINO for video understanding tasks in terms of temporal aspects. We will adjust the wording in our revision to avoid any confusion.
>
> [1] Eyes Wide Shut? Exploring the Visual Shortcomings of Multimodal LLMs
>
> [2] Cambrian-1: A Fully Open, Vision-Centric Exploration of Multimodal LLMs
>
> **W2: Discussion on inference time**
>
> Thanks for raising this concern. Please kindly refer to **general response** for the inference time discussion.
>
> **W3: The proposed approach is transferable to existing MLLMs in a training-free manner**
>
> We applied our compression method to LLaVA-OneVision in a training-free manner and observed performance improvements on EgoSchema, MVBench, and VideoMME, but a performance drop on MLVU. We hypothesize that this is due to the significant reduction in tokens for long video inputs, which differs from the original training distribution of LLaVA-OneVision. While our compression method demonstrates the potential to enhance other MLLMs, further fine-tuning could help bridge the training-inference distribution gap and achieve better results.
>
> | Methods | EgoSchema | MVBench | MLVU | VideoMME |
> | -------- | ------- | ------- | ------- | ------- |
> | LLaVA-OneVision | 60.1 | 56.7 |  64.7 | 58.2 |
> | LLaVA-OneVision + Ours compression | 61.5 | 58.6 | 60.9 | 59.8 |

---

> > ### Comment · Reviewer_kRvH · 2024-11-25
> >
> > Thank you for your response. However, I still have some concerns regarding the runtime of the model and the limited novelty of the paper, so I decide to keep my original score.

---

> ### Author Response · Authors · 2024-11-25
>
> In response to **running time**. We believe there may be a misunderstanding referring to ***“inter-frame similarity calculation”*** and ***“frame-question similarity calculation”*** as ***“dense computations will undoubtedly significantly increase inference time”***.
>
> Below, we provide further clarification and supporting evidence that these two phrases are factually inaccurate.
>
> | (A) Feature extraction Time | Extract DINO feature | Extract SigLIP feature |  DINO similarity | Total Time |
> | -------- | ------- | ------- | ------- | ------- |
> | Naive (SigLIP+DINO) | 22.2 | 20.9 | 0|  43.1 |
> | Ours | 22.2 | 4.32  | 1.05| 27.57 |
>
> If we naively apply the image-based model Cambrian (SigLIP+DINO) for video inference, it requires 22.2 seconds for DINO feature extraction and 20.9 for SigLIP feature extraction for all the video frames. However, by incorporating inter-frame similarity calculation (DINO similarity), we effectively reduce frame-level redundancy, adding only an additional 1.05 seconds to the process. This process significantly decreases the computation required for the remaining frames in subsequent steps for SigLIP feature extraction, from 20.9 (for all frames) to 4.32 (remaining frames), which is 4.83x faster. As a result, inter-frame similarity requires minimal computation time but enables a 1.56x overall speedup in feature extraction time compared to the naive approach.
>
>
> | (B) Compression Component (Query Dependent) | Query | STC | Total |
> | -------- | ------- | ------- | ------- |
> | Inference time (sec) | 0.27 | 0.18 | 0.45 |
>
> The query dependent component takes only a total of 0.45 seconds including 0.27 seconds for the frame-question similarity and  0.18 seconds for STC. As such, the added computation results in only a slight increase in total computation time.
>
>
> | Models  | LLaMA-VID | Chat-UniVi | VideoLLaMA2 | VideoChat2 | LLaVA-OneVision | Ours |
> | -------- | ------- | ------- | ------- | ------- | ------- | ------- |
> | Total execution time (sec) | OOM | 49.06 | 58.62 | 45.22 | 25.84 | 32.96 |
>
>
> When compared with other video baselines, our method demonstrates faster performance compared to the token compression approach of Chat-UniVi, and resampler-based method VideoChat2. Our approach is slightly slower than LLaVA-OneVision, requiring 1.27x the processing time, however, our model improves the performance by 5.2% on average across all the benchmarks shown in Table 1 in the main paper.
>
> **In processing hour-long videos**
>
> One significant advantage of our method is that the DINO-based frame reduction step substantially decreases the computation required for the remaining frames in subsequent steps. Therefore, while **all baseline models encounter OOM issues** when processing video longer than 30 minutes at a 1 FPS sampling rate, our model can process hour-long videos with an inference time of 83.41 seconds, which is **43x times faster** than real time.

---

> > ### Author Response · Authors · 2024-11-25
> >
> > **In response to “limited novelty”**
> >
> > We believe we developed notable contributions that have been thoughtfully developed and carefully integrated to achieve strong performance. We are clarifying here again these contributions hoping we are credited for our efforts.
> >
> > + DINO prior for temporal reduction
> > MoF [1] and Cambrian [2] explore DINOv2’s benefits in low-level image understanding tasks. However, their work does not extend to the video modality, which demands significant training effort and thoughtful design. In contrast, our work takes advantage of DINOv2 similarity to address temporal redundancy (Sec 3.1) in video understanding tasks, offering a novel solution to tackle challenges of long video understanding in MLLMs.
> > + Selective feature reduction based on text query
> > Furthermore, proper design is critical for modeling the spatiotemporal nature of video, as demonstrated in our proposed query-based feature selection (Sec 3.2) adaptively select the frames most relevant to the query at the highest spatial resolution permitted by the model’s capacity.
> > + Spatial token compression by temporal dependency
> > Our spatial token compression method (Section 3.3) compresses visual tokens by leveraging temporal dependencies, effectively reducing redundancy while preserving visual information.
> >
> > [1] Eyes Wide Shut? Exploring the Visual Shortcomings of Multimodal LLMs
> >
> > [2] Cambrian-1: A Fully Open, Vision-Centric Exploration of Multimodal LLMs
> >
> > If you have any further questions or require additional clarification, please do not hesitate to let us know. We would be delighted to engage in a more detailed discussion. Thank you once again for your valuable feedback, which has been instrumental in enhancing the quality of our paper.

---

> ### Author Response · Authors · 2024-12-02
> **Official Comment by Authors**
>
> Dear Reviewer kRvH,
>
> Thank you for your continued feedback. We believe there may have been a misunderstanding regarding the runtime and novelty concerns, as our structured explanations and data suggest a limited impact on runtime and highlight contributions that are unique to our design. We hope this clarifies any remaining doubts and would be happy to discuss further if needed. Thank you again for your time and valuable input.

---

### Official Review · Reviewer_Rw6v · 2024-11-01

**Soundness:** 3
**Presentation:** 4
**Contribution:** 3
**Rating:** 6
**Confidence:** 4

**Summary:**

This paper focuses on the challenge of limited context length in understanding long videos. The authors propose LongVU, which reduces the number of visual tokens across temporal and spatial dimensions. This approach enables the model to handle a large number of frames effectively, achieving strong performance on benchmarks such as VideoMME and MLVU.

**Strengths:**

1. The motivation for reducing the number of visual tokens is clear.

2. The evaluation is thorough, with ablation studies clearly demonstrating the effects of the proposed approaches.

3. The paper is well-written and easy to follow.

**Weaknesses:**

Table 7 indicates that the model performs less effectively on short videos; it would be beneficial to include some analysis of this phenomenon.

This paper is quite comprehensive, and in my view, there are no other significant weaknesses.

**Questions:**

Does the proposed compression approach hurt performance on short videos?

Table 3 lacks results on MVbench, a relatively shorter benchmark. Additionally, Table 7 indicates that the model performs less effectively on short videos. Are there specific limitations causing the proposed method to underperform on short videos? If so, it would be helpful to highlight these limitations.

---

> ### Author Response · Authors · 2024-11-22
> **Response to Reviewer Rw6v**
>
> We sincerely appreciate your positive and constructive comments. Below, we address the potential concern you raised.
>
> **W1: Table 7 indicates that the model performs less effectively on short videos.**
>
> Not quite. In Table 7, our model performs slightly worse than LLaVA-OneVision on the VideoMME short split but surpasses other runner-up models, such as VideoChat2 and LongVA. This is because the VideoMME short video split consists of videos less than 2 minutes, where most questions can be easily answered by analyzing a single frame. Our model outperforms LLaVA-OneVision on average for VideoMME, especially the long video split. In addition, our model outperforms LLaVA-OneVision on other short video benchmarks, such as MVBench (around 16 seconds) by 10.2% and EgoSchema (179.8 seconds) by 7.5%. These benchmarks involve more fine-grained temporal frames, requiring a deeper understanding of temporal dependencies to answer questions accurately.
>
> **Q1: Does the proposed compression approach hurt performance on short videos? Table 3 lacks results on MVbench.**
>
> Thanks for raising this concern. We did not report the MVBench result in Table 3 since it is a very short benchmark and there is only a slight improvement with our spatiotemporal compression on MVBench.
> | Methods | #Tokens | MVBench |
> | -------- | ------- | ------- |
> | DINO | 64 | 66.74 |
> | DINO+Query | 64/144  | 66.82 |
> | DINO+Query+STC | dynamic | 66.88 |
>
> Therefore, there is no negative influence for our spatiotemporal compression approach on short video benchmarks like MVBench.

---

> > ### Comment · Reviewer_Rw6v · 2024-11-25
> >
> > Thank you for your response. The authors have addressed my concerns. I'd like to keep my rating.

---

> > > ### Author Response · Authors · 2024-11-25
> > >
> > > Thanks for your response. We will integrate all your valuable feedback in our revision.

---

### Official Review · Reviewer_jb4v · 2024-11-01

**Soundness:** 4
**Presentation:** 3
**Contribution:** 3
**Rating:** 6
**Confidence:** 4

**Summary:**

This paper introduces LongVU, a novel approach designed to achieve effective long video understanding within the limited context of large language models (LLMs) by compressing the spatio-temporal tokens of videos. The authors provide a detailed discussion on effectively removing redundant temporal frames and spatial tokens. LongVU demonstrates state-of-the-art performance across multiple video understanding benchmarks. The extensive ablation studies convincingly illustrate LongVU's effectiveness in understanding long videos.

**Strengths:**

1. This paper introduces LongVU, which employs two well-designed strategies (Section 3.1, Section 3.3) to reduce redundant temporal frames and spatial tokens, while also proposing the selective retention of high-resolution frames based on queries (Section 3.2). Compared to earlier methods based on frame and token compression [1,2], LongVU demonstrates significant advancements in long video understanding tasks.

2. The authors have conducted comprehensive evaluations, including on the latest long video understanding benchmarks such as VideoMME and MLVU. LongVU achieves state-of-the-art performance in long video understanding tasks compared to recent advanced video MLLMs.

3. The spatial-temporal compression mechanism proposed by the authors has been proven effective for long video tasks, as demonstrated by the ablation study on MLVU in Table 4 and the Needle-in-a-Haystack task experiment in Figure 5.

[1] Song, Enxin, et al. "Moviechat: From dense token to sparse memory for long video understanding." Proceedings of the IEEE/CVF Conference on Computer Vision and Pattern Recognition. 2024.

[2] Li, Yanwei, Chengyao Wang, and Jiaya Jia. "Llama-vid: An image is worth 2 tokens in large language models." European Conference on Computer Vision. Springer, Cham, 2025.

**Weaknesses:**

1. There are some confusing aspects in the methodology section that need clarification:
   - (1) In the Temporal Reduction section (Section 3.1), the authors propose sampling \( T \) frames from \( N \) frames based on DINO similarity. According to the description, \( T \) seems to vary depending on the video. Is there a specific DINO threshold used here? If so, what is the threshold?
   - (2) The proposed method lacks a mechanism to ensure that the tokens fed into the LLM backbone remain within the context length. The authors do not explain how tokens exceeding the LLM context are handled. As shown in Figure 4(b), after token compression (Section 3.2), many video tokens exceed 8K, with some reaching over 15K. Given that the default context length setting is 8K, how are the excess tokens managed?
   - (3) Is the cross-attention module in Section 3.2 trained end-to-end during the video SFT stage?

2. The LongVU 7B model uses Qwen2-7B as the LLM backbone, which has a context length of 32K. Table 3 indicates that their 16K baseline (line 443; Uniform-16K) has a clear advantage over their 8K baseline (line 447; Uniform-8K). Why did the authors ultimately choose 8K for the model setting? Would combining DINO+Query+STC with a 16K or even 32K context length yield better results on hour-level long video tasks? Alternatively, are the authors emphasizing the superiority of DINO+Query+STC under limited context and computational constraints? If so, they should compare the additional computational overhead introduced by DINO+Query+STC.

3. There are some typos in the paper. For example, on line 46, "per framE" has an uppercase "E." In Table 1, some data is misaligned; for instance, the MLVU result for GPT-4o is 64.6, not 66.2.

**Questions:**

1. The authors need to clarify certain ambiguous aspects of their method; please refer to the Weaknesses section for details.

2. The authors should analyze the time and computational efficiency of the proposed compression method. For instance, what proportion of inference time is consumed by calculating DINO similarity between frames and spatial token similarity? Does this lead to significant time overhead during inference?

---

> ### Author Response · Authors · 2024-11-22
> **Response to Reviewer jb4v**
>
> Thank you for providing detailed feedback to enhance the clarity of our paper and for your valuable suggestions to further improve its quality.
>
> **W1: Clarification**
>
> **(1) Is there a specific DINO threshold**
>
> We empirically observe DINO similarities with several videos and set the DINO threshold as 0.83 for all the benchmarks. We ablate the effect of the DINO threshold. The lower the DINO threshold, the fewer frames are reduced. For instance, in MLVU, as the threshold decreases, performance drops due to the retention of more redundant frames within a given context length.
>
> | DINO threshold | EgoSchema | MVBench | MLVU | VideoMME |
> | -------- | ------- | ------- | ------- | ------- |
> | 0.9 | 67.64 | 66.88 | 64.33 | 60.30 |
> | 0.85 | 67.66 | 66.86 | 63.12 | 59.90 |
> | 0.83 (default) | 67.6 | 66.9 | 65.4 | 60.6 |
> | 0.8 | 67.18 | 66.86 | 63.51 | 60.34 |
> | 0.75 | 67.22 | 66.86 | 63.16 | 60.38 |
>
> **(2) The authors do not explain how tokens exceeding the LLM context are handled.**
>
> Thank you for pointing this out. In our implementation, if the number of tokens still exceeds the LLM's context length after all compression steps, we apply forced truncation to the tokens within each sliding window to ensure they fit within the LLM's context window. We will include the explanation in the implementation details of our revision to ensure clarity and prevent any potential confusion.
>
> **(3) Is the cross-attention trained end-to-end during the video SFT stage?**
>
> Yes, the cross-modal query is trained end-to-end during the video SFT stage, enabling it to selectively reduce frame tokens based on the user's query.
>
> **W2: Context length & computation**
>
> For uniformly sampling baselines, due to its sparse sampling nature, taking more frames as input (with increased context length correspondingly) will lead to better performance. But we already argue that uniform sampling is not an optimal solution for video understanding since it has large intervals and loses many visual details.
>
> We chose 8K context length as our setting:
>
> **(i)** Most of the baselines use 8K context length, since the commonly used MLLM is 8K. Hence we conduct for fair comparison.
>
> **(ii)** We have tried our model inference in 12K/16K context length, the result is similar to our 8K setting, which means we can effectively reduce the video redundant tokens by our spatiotemporal compression approach.
>
> | Context length | EgoSchema | MVBench | MLVU | VideoMME |
> | -------- | ------- | ------- | ------- | ------- |
> | 6k | 67.82 | 66.71 | 62.33 | 59.54 |
> | 8k (default)  | 67.6 | 66.9 | 65.4 | 60.6 |
> | 12k | 67.14 | 66.83 | 63.54 | 60.12 |
> | 16k | 67.20 | 66.86 | 64.40 | 60.2 |
>
> **(iii)** Currently, there is a scarcity of long videos with high-quality annotations that can support scaling our methods to longer context lengths. Most videos in our training dataset, VideoChat2-IT, are limited to 3 minutes. Below, we provide an overview of the video lengths for each dataset in VideoChat2-IT.
>
> | TextVR   | MovieChat | YouCook2 | Kinetics-710 | NExTQA  | CLEVRER  | EgoQA    | TGIF     | WebVidQA     | DiDeMo   | ShareGPT4Video |
> |:--------:|:---------:|:--------:|:------------:|:-------:|:--------:|:--------:|:--------:|:------------:|:--------:|:--------------:|
> | Duration      | 15 sec   | 5-20 min  | 5 min    | 10 sec       | 30 sec  | 10 sec   | 1-3 min  | 2-3 sec  | 15-60 sec    | 5-10 sec | 10-20 sec      |
>
> We have also provided the inference computation compared with other baselines in our **general response**.
>
> **W3: Typos**
>
> Thanks for pointing this out. The result for the line of GPT4-o is misaligned, we have fixed the mentioned typos in line 46 and Table 1 in our revised PDF.
>
> **Q2: Inference time consumed by DINO similarity and token similarity**
>
> Please refer the inference time comparison to **general response**. As shown in the table below, the primary computation is from frame feature extraction, which, in real-world applications, can be preprocessed offline. Notably, our proposed compression components contribute only a small portion to the overall inference overhead.
>
> | Component | extract DINO feature | DINO similarity | extract SigLIP feature | Query | STC |
> | -------- | :-------: | :-------: | :-------: | :-------: | :-------: |
> | Inference time (sec) | 22.2 | 1.05 | 4.32 | 0.27 | 0.18 |

---

> > ### Comment · Reviewer_jb4v · 2024-11-25
> >
> > Thank you for your response. The clarifications in the Methods section addressed most of my concerns. I strongly recommend incorporating these into the revised or final version.
> >
> > However, the extraction of DINO features still demands significant inference time, which remains a notable overhead when LongVU processes a new, unseen video.

---

> ### Author Response · Authors · 2024-11-25
>
> Dear Reviewer jb4v,
>
> Thanks for your reply. We have integrated all the results in our revised appendix and we will organize them in our final version.
>
> We would like to clarify that, although the DINO extraction process accounted for the majority of our inference time, it should not be attributed to the computational overhead introduced by our compression method. This is because all models require the first step to extract frame features, and the feature extraction time is comparable across them.
>
> Please kindly notice that in our general response, we provide a comparison of the total inference time compared to baselines.
>
> | Models | LLaMA-VID | Chat-UniVi | VideoLLaMA2 | VideoChat2 | LLaVA-OneVision | Ours |
> | -------- | ------- | ------- | ------- | ------- | ------- | ------- |
> | Inference time (sec) | OOM | 49.06 | 58.62 | 45.22 | 25.84 | 32.96 |
>
> As the table shows, our method demonstrates faster performance compared to the token compression approach of Chat-UniVi, which relies on a KNN-based strategy to merge similar tokens. Furthermore, it is more efficient than the resampler-based method VideoChat2, which compresses video inputs using learnable queries in Q-Former. When compared to methods without compression, such as LLaVA-OneVision, our approach is slightly slower, requiring 1.27x the processing time, however, our model improves the performance by 5.2% on average across all the benchmarks shown in Table 1 in the main paper.
>
> In addition, the feature extraction process is a necessary step that applies to any input videos for any models, regardless of whether the videos are 'new, unseen' as you mentioned.
>
> Please let us know if this clarification resolves the misunderstanding. We are happy to address any additional concerns you may have.
>
> Best,
> Authors

---

> > ### Comment · Reviewer_jb4v · 2024-11-25
> >
> > Thank you for your response. I agree with your clarification. I've increased the soundness score and would like to maintain my rating.

---

> > > ### Author Response · Authors · 2024-11-25
> > >
> > > Thank you for your response! We sincerely appreciate your insightful suggestions for improving the quality of our paper.

---

### Official Review · Reviewer_oDkQ · 2024-11-04

**Soundness:** 3
**Presentation:** 3
**Contribution:** 2
**Rating:** 6
**Confidence:** 4

**Summary:**

This paper aims to address the challenge of token compression in long video understanding by proposing different token sampling strategies in the spatiotemporal dimension: 1) reducing temporal redundancy by leveraging inter-frame feature similarity (DINOv2 and SigLIP); 2) reducing feature redundancy by utilizing cross-modal similarity; and 3) reducing redundant tokens through element-wise cosine similarity within spatial windows.
Through these effective and intuitive strategies, this research is capable of processing 1-hour-long videos within a common 8k context length. Additionally, this paper demonstrates superior numerical performance compared to other long video understanding methods such as VideoChat2 and LLaVA-OneVision.

**Strengths:**

1. This paper is competitive in long video understanding performance; for instance, its performance on VideoMME long videos surpasses that of LLaVA-OneVision by 12.8 points.
2. In smaller model sizes (LLaMA3.2-3B), LongVU also outperforms existing methods.
3. This method essentially proposes a token compression strategy that helps reduce token redundancy in long videos. It is a general approach that can be applied across different multi-modal large language models (MLLMs).

**Weaknesses:**

1. This paper is engineering-oriented and lacks some innovation. For instance, the combination of SigLIP and DINOv2 is based on the approach used in Cambrian. Additionally, the ability of DINOv2 features to more intuitively reflect low-level visual characteristics is evident, and this has also been compared with CLIP-based methods in the original paper.
2. In the comparative experiments, I believe that including the inference time for processing videos of the same length would contribute to a fairer comparison. This is because the token compression strategy employed in this paper inevitably introduces some preprocessing steps, which in turn adds more inference time.

**Questions:**

1. Is the sliding window size \( K = 8 \) an empirical value in spatial token compression (STC)?
2. Are there any analyses of failed examples?

---

> ### Author Response · Authors · 2024-11-22
> **Response to Reviewer oDkQ**
>
> We sincerely thank you for your thorough review of our paper. Your valuable suggestions are instrumental in enhancing the quality and clarity of our work, and we deeply appreciate your thoughtful feedback.
>
> **W1: Lacks of innovation, e.g., DINOv2 has been explored**
>
> Although Cambrian explores DINOv2 benefits in low-level image understanding tasks, however, we are the first to utilize DINOv2 similarity to reduce temporal redundancy (Sec 3.1) in video understanding tasks, different from previous literatures that only work on image modality. Furthermore, proper design is critical for modeling the spatiotemporal nature of video, as demonstrated in our proposed query-based feature selection (Sec 3.2) adaptively select the frames most relevant to the query at the highest spatial resolution permitted by the model’s capacity and spatial token compression (Sec 3.3) where visual tokens are compressed based on temporal dependency.
>
> **W2: Inference time comparison**
>
> Please kindly refer the inference time comparison to **general response**. Our method demonstrates more efficient performance compared to the token compression approach of Chat-UniVi, and resampler-based method VideoChat2. Compared to non-compression methods like LLaVA-OneVision, our approach is 1.27x slower but delivers a 5.2% average performance improvement across all benchmarks in Table 1 of the main paper.
>
>
> **Q1: Sliding window size of K=8**
>
> $K$ represents the size of the sliding window, a hyperparameter used to divide the frame sequence into chunks for spatial token compression. Below, we show the ablation results for different values of $K$. Adjusting the sliding window size will affect the MLVU performance, while having minimal impact for other benchmarks. In STC, each subsequent frame is compared with a first frame as anchor within the sliding window. We assume that a larger context window can weaken the strong temporal dependency when the subsequent frame is far from the initial frame in the window.
>
> | Sliding window $K$ | EgoSchema | MVBench | MLVU | VideoMME |
> | -------- | ------- | ------- | ------- | ------- |
> | 4 | 67.38 | 66.86 | 63.74 | 60.45 |
> | 8 (default) | 67.6 | 66.9 | 65.4 | 60.6 |
> | 16 | 67.22 | 66.86 | 62.18 | 60.42 |
> | 32 | 67.20 | 66.86 | 60.69 | 60.82 |
>
> **Q2: Analysis of failed examples**
>
> Our method spatiotemporally reduces video frames/tokens and concatenates tokens all together to form the overall video representation. However, this approach does not encode the temporal location of individual frames. While we experimented with frame-level positional embeddings to alleviate this drawback, the model still struggles with tasks like temporal grounding, meaningly identifying the precise start and end times of events.

---

> > ### Comment · Reviewer_oDkQ · 2024-11-25
> >
> > Thank you for your response. Your clarification  has addressed most of my concerns. I will improve my score.
> >
> > Since the methods in your paper are simple and intuitive, such as using DINOv2 features to eliminate video redundancy, it is necessary to emphasize the similarities and differences between your approach and previous methods, especially those related to long video compression.
> >
> > Additionally, as you mentioned, LongVU selectively compresses spatiotemporal tokens, which makes it difficult to perceive the positional information between frames. Discussing these limitations and pointing out possible solutions would also be meaningful.

---

> > > ### Author Response · Authors · 2024-11-25
> > >
> > > Dear Reviewer oDkQ,
> > >
> > > Thank you for your constructive feedback. We’re encouraged to see your positive response to our earlier comments, and we’d like to share our discussion based on your valuable advices.
> > >
> > > **Related work discussion**
> > >
> > >  We have included all known long video compression methods, including Video-ChatGPT [1], LLaMA-VID [2], MovieChat [3], MA-LMM [4], and Chat-UniVi [5], to the best of our knowledge.
> > >
> > > Video-ChatGPT [1] employs pooling-based methods to reduce tokens, while MA-LMM [4] and MovieChat [3] maintain a memory to facilitate long video understanding. LLaMA-VID [2] compresses videos based on user queries encoded by a lightweight language model, and Chat-UniVi [5] adopts KNN-based clustering techniques to effectively merge tokens.
> > >
> > > **Similarities & Differences**
> > >
> > > In the context of frame similarity, MovieChat [3] reduces frames based on CLIP feature similarity. However, our findings indicate that DINO features are more effective for capturing pixel-level differences between video frames.
> > >
> > > In terms of addressing long videos based on text queries, LLaMA-VID [2] incorporates an additional text decoder to embed text queries and enable cross-attention with frame features. Goldfish [6], on the other hand, generates captions for each short clip, embeds them into text vectors using another LLM, and computes cosine similarity between the text query embedding and the clip caption embeddings within a purely linguistic space. In contrast, we selectively reduce frame tokens based on the user's text queries in MLLM’s aligned space.
> > >
> > > **Limitation Discussion**
> > >
> > > We have updated the limitations related to the positional information between frames. We think that a well-designed frame-level positional embedding could help address this issue. Alternatively, explicitly adding <frame_i> text to demonstrate the timestamp of each frame or overlaying visual text on the frames to indicate their timestamps could also be a potential solution. We will explore and improve in this direction in future work.
> > >
> > > [1] Video-ChatGPT: Towards detailed video understanding via large vision and language models, ACL 2024
> > >
> > > [2] LLaMA-VID: An Image is Worth 2 Tokens in Large Language Models, ECCV 2024
> > >
> > > [3] MovieChat: From Dense Token to Sparse Memory for Long Video Understanding, CVPR 2024
> > >
> > > [4] MA-LMM: Memory-Augmented Large Multimodal Model for Long-Term Video Understanding, CVPR 2024
> > >
> > > [5] Chat-UniVi: Unified Visual Representation Empowers Large Language Models with Image and Video Understanding, CVPR 2024
> > >
> > > [6] Goldfish: Vision-Language Understanding of Arbitrarily Long Videos, ECCV 2024
> > >
> > > We will organize all the comments and integrate in our final version. Thanks again for your valuable suggestions in improving the quality of the paper.

---

### Official Review · Reviewer_LGhK · 2024-11-04

**Soundness:** 3
**Presentation:** 3
**Contribution:** 2
**Rating:** 6
**Confidence:** 4

**Summary:**

The paper presents LongVU, a spatiotemporal adaptive compression mechanism designed to handle long video sequences within the limited context length constraints of current multimodal large language models (MLLMs). The authors propose leveraging cross-modal query and inter-frame dependencies to reduce temporal and spatial redundancy in videos effectively. The method demonstrates promising results across various video understanding benchmarks, particularly for hour-long videos.

**Strengths:**

1. The concept of spatiotemporal adaptive compression to address the challenge of long video understanding is timely, given the increasing interest in multimodal learning and the limitations of current MLLMs.
2. The paper provides extensive experimental validation, showing significant improvements over existing methods on multiple video understanding benchmarks, which is a strong empirical demonstration of the proposed method's effectiveness.
3. The use of DINOv2 features for temporal reduction and the cross-modal query mechanism for selective frame feature reduction are well-motivated and technically sound.

**Weaknesses:**

1. While the paper discusses the effectiveness of LongVU, a deeper analysis into the computational complexity and scalability, especially regarding the trade-offs with comprasion ratio and context length, could be beneficial.
2. The paper focuses on video-language understanding. It would be useful to understand how well these techniques generalize to other modalities (e.g. image).
3. More ablation studies on the components of the spatiotemporal compression strategy could provide further insights into the contribution of each component to the overall performance.
4. The query-based comparison seems limited to single-turn dialogue scenarios. The authors should consider how LongVU would handle multi-turn interactions, where the context and relevance of frames may change dynamically across turns. Addressing this could significantly expand the applicability of LongVU.
5. The paper missed some relevant works, such as "LGDN: Language-guided denoising network for video-language modeling" (NeurIPS 2022).

**Questions:**

Please see weaknesses.

---

> ### Author Response · Authors · 2024-11-22
> **Response to Reviewer LGhK Part [1/2]**
>
> We greatly appreciate your valuable comments. Below, we address your questions, hoping to resolve your concerns.
>
> ---
>
> **W1: Computational complexity and scalability, trade-offs with compression ratio and context length**
>
> Thanks for your concerns. We have provided the inference complexity comparison in the general response. Our proposed compression method reduces computational overhead compared to other approaches, such as Chat-UniVi and VideoChat2.
>
> We conducted an ablation study to compare the impact of different context lengths on our method. By default, we use an 8K context length, as it aligns with most baselines and is the standard for commonly used MLLMs, ensuring a fair comparison.
> When reducing the context length to 6K, performance suffers due to the restricted context window. Conversely, testing our model with extended context lengths of 12K and 16K yielded results comparable to the 8K setting. This demonstrates that our spatiotemporal compression approach effectively minimizes redundant video tokens, maintaining performance even with longer contexts.
> In addition, currently, there is a scarcity of long videos with high-quality annotations that can support scaling our methods to longer context lengths. Most videos in our training dataset, VideoChat2-IT, are limited to 3 minutes.
>
> | Context length | EgoSchema | MVBench | MLVU | VideoMME |
> | -------- | ------- | ------- | ------- | ------- |
> | 6k | 67.82 | 66.71 | 62.33 | 59.54 |
> | 8k (default)  | 67.6 | 66.9 | 65.4 | 60.6 |
> | 12k | 67.14 | 66.83 | 63.54 | 60.12 |
> | 16k | 67.20 | 66.86 | 64.40 | 60.20 |
>
> We investigate the impact of DINO threshold selection on performance. A lower DINO threshold results in fewer frames being discarded (lower compression rate). For example, in MLVU, decreasing the threshold leads to performance drop as more redundant frames are retained in a given context length.
> | DINO threshold | EgoSchema | MVBench | MLVU | VideoMME |
> | -------- | ------- | ------- | ------- | ------- |
> | 0.9 | 67.64 | 66.88 | 64.33 | 60.30 |
> | 0.85 | 67.66 | 66.86 | 63.12 | 59.90 |
> | 0.83 (default) | 67.6 | 66.9 | 65.4 | 60.6 |
> | 0.8 | 67.18 | 66.86 | 63.51 | 60.34 |
> | 0.75 | 67.22 | 66.86 | 63.16 | 60.38 |
>
> We analyze the impact of STC threshold selection on performance and observe a trade-off with the compression rate. Increasing the threshold, meaningly reducing the compression rate, leads to a drop in performance, particularly on the long-video benchmarks MLVU and VideoMME. Conversely, lowering the threshold results in more aggressive compression, which also negatively impacts performance.
>
> | STC threshold $\theta$ | EgoSchema | MVBench | MLVU | VideoMME |
> | -------- | ------- | ------- | ------- | ------- |
> | 0.85 | 67.56 | 66.88 | 64.00 | 59.98 |
> | 0.8 | 67.3 | 66.86 | 63.51 | 59.83 |
> | 0.75 (default) | 67.6 | 66.9 | 65.4 | 60.6 |
> | 0.7 | 67.5 | 66.86 | 64.03 | 60.27 |
> | 0.65 | 67.42 | 66.86 | 63.91 | 60.34 |

---

> > ### Author Response · Authors · 2024-11-22
> > **Response to Reviewer LGhK Part [2/2]**
> >
> > **W2: How the model generalizes in image modality**
> >
> > Our spatiotemporal compression approach is specifically designed to address the challenges of long video understanding tasks. Our first component, frame-level reduction can be only applied for video, inapplicable for image modality. Moreover, our spatial compression also relies on temporal dependencies across frames. As shown in Table 10 (appendix), our image-pretrained model demonstrates good image understanding capabilities. However, after fine-tuning on video tasks, a modality gap leads to a decline in image performance. Interestingly, when the video SFT model is further fine-tuned on the image data from the pretraining stage, the image performance is effectively restored. This indicates that our approach to video understanding can also enhance image understanding capabilities. The recovered image understanding performance shown in below table is comparable to MLLM baselines, such as Cambrian-1 [1] and LLaVA-NeXT [2], which primarily focus on image understanding and operate with a similar number of image tokens.
> >
> > | Model   |  GQA | MMVP | POPE | RealWorldQA |
> > | -------- | ------- | ------- | ------- | ------- |
> > | Cambrian-1 | 64.04 | 32.67 | - | 58.95 |
> > | LLaVA-NeXT | 65.2 | 38.7 | 86.5 | 64.2 |
> > | Ours (Before video SFT)  |  62.26  | 51.33 | 86.65 | 61.06 |
> > | Ours (After video SFT) |  60.83  | 32.00 | 81.23 | 47.65  |
> > | Ours (After video SFT + image SFT)  |  69.65  | 52.66 | 86.22 | 62.10|
> >
> > [1] Cambrian-1: A fully open, vision-centric exploration of multimodal llms, Neurips 2024
> >
> > [2] LLaVA-NeXT: Improved reasoning, OCR, and world knowledge, technical report 2024
> >
> > ---
> >
> > **W3: Each component of the spatiotemporal compression strategy**
> >
> > We have demonstrated in Tables 3 and 4 the step-by-step performance improvements achieved by incrementally adding each component, i.e, DINO, Query, and STC. We also included additional ablations for the DINO threshold and drop threshold as mentioned in the response to W1.
> >
> > ---
> >
> > **W4: Limited to single-turn dialogue scenarios**
> >
> > The query-based selective feature compression can be easily adapted for multi-turn conversation settings, by locating the current-turn’s user query by splitting multiple turns by <|im_start|>user (Qwen chat template) or <|start_header_id|>user (Llama3.2 chat template) corresponding token. To the best of our knowledge, there are currently no publicly available benchmarks for multi-round video question answering. We appreciate your suggestion and would like to explore more in the future.
> >
> > ---
> >
> > **W5: Missing relevant works like LGDN**
> >
> > Thanks for sharing the work. We included several related works [1-3] in line 149 and line 158  in our revised PDF.
> >
> > [1] LGDN: Language-guided denoising network for video-language modeling, Neurips 2024
> >
> > [2] LongVILA: Scaling Long-Context Visual Language Models for Long Videos, arXiv:2408.10188
> >
> > [3] MA-LMM: Memory-Augmented Large Multimodal Model for Long-Term Video Understanding, CVPR 2024

---

> > > ### Author Response · Authors · 2024-12-02
> > >
> > > Dear reviewer LGhK,
> > >
> > > We really appreciate your efforts to help improve this paper. We have carefully addressed all the mentioned concerns:
> > > + Discussion on the inference time, including comparisons with baseline models and a detailed analysis of each proposed;
> > > + Additional ablation experiments on context lengths, compression threshold;
> > > + Model performance on image modality;
> > > + Discuss the multi-turn scenarios;
> > > + Add more related works.
> > >
> > > We kindly ask if our response has effectively addressed your concerns, and we are eager to hear your valuable feedback.
> > >
> > > Best,
> > >
> > > Authors

---

### Author Response · Authors · 2024-11-22
**General Response**

We thank the reviewers for their insightful feedback. We are encouraged by their recognition of the following:
+ Our proposed spatiotemporal adaptive compression demonstrates **effectiveness** in understanding long videos (@LGhK, @jb4v), and also scales well in **smaller models** (@oDkQ).
+ The **motivation is clear**: to tackle the challenge of long video understanding in current MLLMs (@LGhK, @Rw6v).
+ The paper is **well-written** (@Rw6v, @kRvH), **technically sound** (@LGhK), and supported by **comprehensive experiments** across extensive evaluations (@jb4v, @Rw6v).

We address common questions raised by the reviewers here.

 **Model inference time comparison**

To evaluate the computational overhead introduced by our proposed spatiotemporal compression approach, we compare it with various baselines using input videos of the same length (20 minutes) sampled at 1 FPS. The experiments were conducted on an A100 GPU with 80 GB memory. LLaMA-VID encounters a CUDA out-of-memory (OOM) issue when processing 20-minute videos as input. As the table shows below, our method demonstrates faster performance compared to the token compression approach of Chat-UniVi, which relies on a KNN-based strategy to merge similar tokens. Furthermore, it is more efficient than the resampler-based method VideoChat2, which compresses video inputs using learnable queries in Q-Former. When compared to methods without compression, such as LLaVA-OneVision, our approach is slightly slower, requiring 1.27x the processing time, however, our model improves the performance by 5.2% on average across all the benchmarks shown in Table 1 in the main paper.

| Models | LLaMA-VID | Chat-UniVi | VideoLLaMA2 | VideoChat2 | LLaVA-OneVision | Ours |
| -------- | :-------: | :-------: | :-------: | :-------: | :-------: | :-------: |
| Inference time (sec) | OOM | 49.06 | 58.62 | 45.22 | 25.84 | 32.96 |

**Inference time of each compression component**

We begin by using the DINOv2 vision encoder to extract features from all frames and then reduce redundant frames based on DINO feature similarity. After this reduction, the remaining frames are processed using SigLIP. We use input videos of 20 minutes sampled at 1 FPS and the experiments are conducted on an A100 GPU with 80 GB memory. As shown in the table below, the primary computation lies in frame feature extraction, which, in real-world applications, can be preprocessed offline. Notably, our proposed compression components contribute only a small portion to the overall inference overhead.
| Component | Extract all frame features | **DINO similarity** | Extract remaining feature for SigLIP | **Query** | **STC** | Total |
| -------- | :-------: | :-------: | :-------: | :-------: | :-------: | :-------: |
| Inference time (sec) | 22.2 | 1.05 | 4.32 | 0.27 | 0.18 | 32.96 |

One significant advantage of our method is that the DINO-based frame reduction step substantially decreases the computation required for the remaining frames in subsequent steps. Therefore, while most baseline models encounter OOM issues when processing hour-long videos at a 1 FPS sampling rate, our model can process hour-long videos with an inference time of 83.41 seconds.

We respond to specific queries in each reviewer's section. We will incorporate all constructive feedback into our revision.

---

### Meta-Review · Area_Chair_Jyu9 · 2024-12-19

**Metareview:**

This paper proposes LongVU, a spatiotemporal adaptive compression mechanism for long video-language understanding. It aims to address the challenge of limited context length in current multimodal large language models (MLLMs) by reducing video tokens while preserving visual details. Experiments are conducted on multiple benchmarks, including VideoMME and MLVU.

The main strengths are: 1) well-motivated and technically sound model design (Spatiotemporal compression reduces tokens and retains key details) and 2) good performance.

The main weaknesses are: 1) limited novelty (the combination of SigLIP and DINOv2 is based on the approach used in Cambrian, and several similar works already exist [1-2]) and 2) missing technique details and explanation of experimental settings.
[1] Eyes Wide Shut? Exploring the Visual Shortcomings of Multimodal LLMs
[2] Cambrian-1: A Fully Open, Vision-Centric Exploration of Multimodal LLMs

Since the technique novelty of this paper is limited (recognized by the reviewers oDkQ, and kRvH), the AC does not recommend acceptance at this conference. The authors are encouraged to address these concerns for future submissions.

**Additional Comments On Reviewer Discussion:**

After the discussion, most of the concerns on technique details and experimental settings have been addressed, but reviewers oDkQ, and kRvH still have concerns on the technique novelty of this paper.

---

### Decision · Program_Chairs · 2025-01-22

Reject